# Learning N:M Fine-grained Structured Sparse Neural Networks From Scratch

**Aojun Zhou**[1,2][*]**, Yukun Ma**[3][*]**, Junnan Zhu**[4]**, Jianbo Liu**[2]**, Zhijie Zhang**[1]**, Kun Yuan**[1]
**Wenxiu Sun**[1] **Hongsheng Li**[2]
[1]SenseTime Research, [2]CUHK-Sensetime Joint Lab, CUHK, [3]Northwestern University, [4]NLPR, CASIA
aojunzhou@gmail.com  gmayukun@gmail.com  hsli@ee.cuhk.edu.hk

## Abstract

Sparsity in Deep Neural Networks (DNNs) has been widely studied to compress and accelerate the models on resource-constrained environments. It can be generally categorized into unstructured fine-grained sparsity that zeroes out multiple individual weights distributed across the neural network, and structured coarse-grained sparsity which prunes blocks of sub-networks of a neural network. Fine-grained sparsity can achieve a high compression ratio but is not hardware friendly and hence receives limited speed gains. On the other hand, coarse-grained sparsity cannot concurrently achieve both apparent acceleration on modern GPUs and decent performance. In this paper, we are the first to study training from scratch an *N:M* fine-grained structured sparse network, which can maintain the advantages of both unstructured fine-grained sparsity and structured coarse-grained sparsity simultaneously on specifically designed GPUs. Specifically, a $2:4$ sparse network could achieve $\mathbf{2}\times$ speed-up without performance drop on Nvidia A100 GPUs. Furthermore, we propose a novel and effective ingredient, sparse-refined straight-through estimator (SR-STE), to alleviate the negative influence of the approximated gradients computed by vanilla STE during optimization. We also define a metric, Sparse Architecture Divergence (SAD), to measure the sparse network's topology change during the training process. Finally, We justify SR-STE's advantages with SAD and demonstrate the effectiveness of SR-STE by performing comprehensive experiments on various tasks. Source codes and models are available at https://github.com/NM-sparsity/NM-sparsity.

## 1 Introduction

Deep neural networks (DNNs) have shown promising performances on various tasks including computer vision, natural language processing, speech recognition, etc. However, a DNN usually comes with a large number of learnable parameters, ranging from millions of to even billions of (*e.g.*, GPT-3 (Brown et al., 2020)), making the DNN model burdensome and difficult to be applied to real-world deployments. Therefore, researchers began to investigate how to speed up and compress DNNs via various methods such as knowledge distillation (Hinton et al., 2015), quantization (Jacob et al., 2018; Zhou et al., 2017), designing efficient model architectures (Howard et al., 2017), and structured sparsity (Wen et al., 2016; Li et al., 2016).

In this paper, we focus on the problem of sparsifying DNNs. Sparsity in DNNs can be categorized into unstructured sparsity and structured sparsity. Unstructured sparsity prunes individual weights at any location, which is fine-grained and can achieve extremely high compression ratio (Han et al., 2015; Guo et al., 2016). However, unstructured sparsity struggles to take advantage of vector-processing architectures, which increases latency due to dependent sequences of reads (Nvidia, 2020). Compared with unstructured sparsity, structured sparsity is more friendly to hardware, especially for block pruning (Wang et al., 2019), kernel shape sparsity (Tan et al., 2020) or channel and filter pruning (Li et al., 2016; Wen et al., 2016). Although structured sparsity can speed up DNNs on commodity hardware, it hurts model performance more significantly than unstructured fine-grained sparsity. For example, ResNet-50 network generated by unstructured pruning can achieve a $5.96\times$ compression ratio, with the same accuracy as the original network, but it can only achieve $1\times$ com-

---

[*]The first two authors equally contribute to this paper.

pression in the case of structured sparsity (Renda et al., 2020). Therefore, how to combine the unstructured sparsity and structured sparsity to accelerate DNNs on modern hardware (*e.g.*, GPU) becomes a challenging yet valuable problem. Recently, Nvidia Ampere A100 is equipped with the **Sparse Tensor Cores** to accelerate 2:4 structured fine-grained sparsity. Here, *N:M* sparsity indicates the sparsity of DNNs in which only $N$ weights are non-zero for every continuous $M$ weights. To the best of our knowledge, A100 is the first commodity sparse hardware, where the sparse tensor core can support several common operations including linear, convolutional, recurrent cells, transformer blocks, etc. Specifically, suppose a typical matrix multiplication $\mathcal{X} \times \mathcal{W}$ in DNNs, $\mathcal{X}$ and $\mathcal{W}$ denote input tensor and parameter tensor respectively. The Dense Tensor Cores implement $\mathcal{X}_{16 \times 32} \times \mathcal{W}_{32 \times 8}$ matrix multiplication by 2 cycles while the Sparse Tensor Cores only need 1 cycle if the parameter tensor $\mathcal{W}$ satisfies the 2:4 structured sparse pattern.

Nvidia has proposed an ASP[1] (APEX's Automatic Sparsity) solution (Nvidia, 2020) to sparsify a dense neural network to satisfy the 2:4 fine-grained structured sparsity requirement. The recipe contains three steps: (1) training a dense network until converge; (2) pruning for *2:4* sparsity with magnitude-based single-shot pruning; (3) repeating the original training procedure. However, ASP is computationally expensive since it requires training the full dense models from scratch and fine-tuning again. Therefore, we still lack a simple recipe to obtain a structured sparse DNN model consistent with the dense network without extra fine-tuning.

This paper addresses this question: *Can we design a simple yet universal recipe to learn $N:M$ sparse neural networks from scratch in an efficient way*?

It is difficult to find the optimal sparse architecture (connections) and optimal parameters (Evci et al., 2019b) simultaneously during training sparse CNNs and Transformers although SET-MLP could easily outperform dense MLP (Bourgin et al., 2019). There are two schemes to obtain such sparse models. One is a two-stage scheme, which discovers a sparse neural architecture by pruning a well-trained dense network and then uses the same or even greater computational resources to retrain the sparse models (Nvidia, 2020; Evci et al., 2019b; Han et al., 2015; Frankle & Carbin, 2018). The other is a one-stage scheme, which adopts the dynamic method to alternatively optimize parameters and prunes network architectures based on different criteria (Bellec et al., 2017; Mocanu et al., 2018; Mostafa & Wang, 2019; Evci et al., 2019b; Kusupati et al., 2020; Dettmers & Zettlemoyer, 2019). Compared with the two-stage scheme, the one-stage scheme can save training time and cost however usually obtains lower performance.

To overcome the aforementioned trade-off between training cost and performance, we present a simple yet effective framework to train sparse neural networks from scratch. Specifically, we employ the magnitude-based pruning method (Renda et al., 2020; Gale et al., 2019) during the forward process. Considering that the pruning operation is a non-differentiable operator (a similar dilemma in model quantization (Courbariaux et al., 2016)), we extend the widely used Straight-through Estimator (STE) (Bengio et al., 2013) in model quantization to aid sparse neural network's back-propagation. However, perturbations are introduced during the back-propagation (Yin et al., 2019; Bengio et al., 2013). Hence we define Sparse Architecture Divergence (SAD) to further analyze $N:M$ sparse networks trained by STE methods so that we can identify the impact of perturbations on sparse neural networks training. Based on SAD analysis, to alleviate the negative impact, we propose a sparse-refined term mitigating the approximated gradients' influence.

We also compare the performance of neural networks with different granularities of fine-grained structured sparsity (*i.e.*, 1:4, 2:4, 2:8, 4:8) and conduct thorough experiments on several typical deep neural networks with different $N:M$ sparsity levels, covering image classification, detection, segmentation, optical flow estimation, and machine translation. Experimental results have shown that the models with our proposed structured sparsity can achieve neglectful performance drop and can even sometimes outperform the dense model.

The main contributions of this paper are summarized as three-fold. **(1)** To the best of our knowledge, this is the first systematic study into training $N:M$ structured sparse neural networks from scratch without performance drop. The $N:M$ structured sparsity is a missing yet promising ingredient in model acceleration, which can be a valuable supplement with various compression methods. **(2)** We extend STE to tackle the problem of training $N:M$ sparse neural networks. To alleviate the limitations of STE on sparsifying networks, we propose a sparse refined term to enhance the effectiveness

---

[1] https://github.com/NVIDIA/apex/tree/master/apex/contrib/sparsity

on training the sparse neural networks from scratch. **(3)** We conduct extensive experiments on various tasks with $N$:$M$ fine-grained sparse nets, and provide benchmarks for $N$:$M$ sparse net training to facilitate co-development of related software and hardware design.

## 2 RELATED WORK

**Unstructured and Structured Sparsity.** Sparsity of DNNs is a promising direction to compress and accelerate a deep learning model. Among all sparsity types, unstructured sparsity can achieve a significantly high compression ratios (*e.g.* $13\times$ (Han et al., 2015) and $108\times$ (Guo et al., 2016)) while ensuring decent accuracy by pruning. Many different pruning criterions and pruning methods are proposed for unstructured sparsity, *e.g.*, magnitude-based pruning (Han et al., 2015; Frankle & Carbin, 2018), Hessian based heuristics (LeCun et al., 1990), and pruning with connection sensitivity (Lee et al., 2018). However, unstructured sparsity's ability to accelerate is highly limited since it takes a lot of overhead to store the irregular non-zero index matrix. On the other hand, Wen et al. (2016) introduces the structural sparsity to speed up deep models on GPUs. Existing structural sparsity contains filter-wise sparsity (Li et al., 2016), channel-wise sparsity (Li et al., 2016), filter-shape-wise sparsity. Different from existing sparsity patterns (fine-grained unstructured sparsity and coarse-grained structured sparsity), this paper presents an $N$:$M$ fine-grained structured sparsity, a sparsity type that has both high efficiency and lossless performance.

**One-stage and two-stage methods.** There are mainly two types of techniques to obtain a sparse neural network, one-stage methods and two-stage ones. The two-stage method first prunes a trained dense neural network and then retrains fixed sparse network to recover its performance. Typical two-stage methods include single-shot pruning (Lee et al., 2018) and iterative pruning (Han et al., 2015; Guo et al., 2016). Later, the lottery ticket hypothesis (Frankle & Carbin, 2018) shows that the sparse sub-network (winning tickets) can be trained from scratch with the same initialization while the winning tickets are discovered by dense training. Deep-Rewiring (Bellec et al., 2017), on the other hand, is a typical one-stage method, which takes a Bayesian perspective and samples sparse network connections from a posterior, however is computationally expensive and challenging to be applied to large-scale tasks. Sparse Evolutionary Training (Mocanu et al., 2018)(SET) is proposed as a simpler scheme where weights are pruned according to the standard magnitude criterion used in pruning and growing connections in random locations. Dettmers & Zettlemoyer (2019) uses the momentum of each parameter as the criterion for growing weights and receives an improvement in test accuracy. GMP (Gale et al., 2019) trains the unstructured sparse net using variational dropout and $l_0$ regularization from scratch, and shows that unstructured sparse architectures learned through pruning cannot be trained from scratch to have the same testing performance as dense models do. Recently proposed state-of-the-art method STR (Kusupati et al., 2020) introduces pruning learnable thresholds to obtain a non-uniform sparse network. RigL (Evci et al., 2019a) uses the magnitude-based method to prune and the periodic dense gradients to regrow connection. However, compared with training dense neural networks from scratch, to achieve the same performance, RigL needs $5\times$ more training time.The most closely related work to ours may be DNW(Wortsman et al., 2019) which uses a fully dense gradient in the backward run to discover optimal wiring on the fly.

## 3 METHOD

### 3.1 *N:M* FINE-GRAINED STRUCTURED SPARSITY

Here we define the problem of training a neural network with $N$:$M$ fine-grained structured sparsity. A neural network with $N$:$M$ sparsity satisfies that, in each group of $M$ consecutive weights of the network, there are at most $N$ weights have non-zero values. Fig. 1 illustrates a 2:4 sparse network.

Generally, our objective is to train an $N$:$M$ sparse neural network as

$$\min_{S(\mathcal{W},N,M)} \mathcal{L}(\mathcal{W};\mathcal{D}), \tag{1}$$

where $\mathcal{D}$ denotes the observed data, $\mathcal{L}$ represents the loss function, $\mathcal{W} = \{\mathcal{W}^l : 0 < l \leqq L\}$ indicates the parameters of an $L$-layer neural network, and $S(\mathcal{W}, N, M)$ is the $N$:$M$ sparse neural network parameters.

### 3.2 STRAIGHT-THROUGH ESTIMATOR(STE) ON TRAINING *N:M* SPARSE NETWORKS

A straightforward solution for training an $N$:$M$ sparsity network is to simply extend Straight-through Estimator (STE) (Bengio et al., 2013) to perform online magnitude-based pruning and

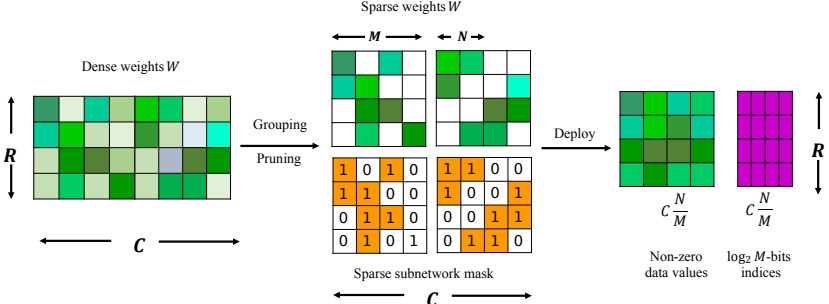

Figure 1: Illustration of achieving $N{:}M$ structure sparsity. (Left) In a weight matrix of 2:4 sparse neural network, whose shape is $R \times C$ (*e.g.*, $R =$ output_channels and $C =$ input_channels in a linear layer), at least two entries would be zero in each group of 4 consecutive weights. (Middle & Right) The process that the original matrix is compressed, which enables processing of the matrix to be further accelerated by designated processing units (*e.g.*, Nvidia A100).

sparse parameter updating, which is depicted in Fig. 2(a). STE is widely used in model quantization (Rastegari et al., 2016), since the quantized function is non-differentiable without STE and the networks optimized with STE has decent performance under careful settings (Yin et al., 2019). In STE, a dense network is maintained during the training process. During the forward pass, we project the dense weights $\mathcal{W}$ into sparse weights $\widetilde{\mathcal{W}} = S(\mathcal{W}, N, M)$ satisfying $N{:}M$ sparsity. Let $\mathbf{w} \subset \mathcal{W}$ be a group of consecutive $M$ parameters in $\mathcal{W}$ and $\widetilde{\mathbf{w}} \subset \widetilde{\mathcal{W}}$ be the corresponding group in $\widetilde{\mathcal{W}}$. The projection of $\mathbf{w}$ can be formulated as:

$$\widetilde{\mathbf{w}}_i = \begin{cases} \mathbf{w}_i & \text{if } |\mathbf{w}_i| \geqslant \xi \\ 0 & \text{if } |\mathbf{w}_i| < \xi \end{cases}, \text{for } i = 1, 2, \ldots, M \tag{2}$$

where $\xi$ is the $N$-th largest value in $\mathbf{w} = \{|\mathbf{w}_1|, |\mathbf{w}_2|, \ldots, |\mathbf{w}_M|\}$. Intuitively, this projection function $S(\cdot)$ produces sparse parameters $\widetilde{\mathcal{W}}$ by setting $N$ parameters that have the least significant absolute values to zero in each consecutive $M$-parameter group, while keeping the other parameters the same as before. The computation of an $N{:}M$ sparse sub-network on-the-fly in the forward pass is illustrated in Fig. 1.

The projection function $S(\cdot)$, which is non-differentiable during back-propagation, generates the $N{:}M$ sparse sub-network on the fly. To get gradients during back-propagation, STE computes the gradients of the sub-network $g(\widetilde{\mathcal{W}}) = \nabla_{\widetilde{\mathcal{W}}} \mathcal{L}(\widetilde{\mathcal{W}}; \mathcal{D})$ based on the sparse sub-network $\widetilde{\mathcal{W}}$, which can be directly back-projected to the dense network as the approximated gradients of the dense parameters. The approximated parameter update rule for the dense network (see Fig. 2(a) in Appendix) can be formulated as

$$\mathcal{W}_{t+1} \leftarrow \mathcal{W}_t - \gamma_t g(\widetilde{\mathcal{W}}_t), \tag{3}$$

where $\mathcal{W}_t$ represents dense parameters at iteration $t$ and $\gamma_t$ indicates the learning rate.

### 3.2.1 ANALYSIS OF DYNAMIC PRUNING USING STE

To validate the performance of STE on N:M sparse networks, we perform a pilot experiment with STE. The results are shown in Fig. 3(a). From Fig. 3(a), $N{:}M$ neural network trained with the above mentioned STE shows significant performance drop compared with the dense network. We conjecture that this drop results from unstable neural architecture updates caused by the approximated gradients of the dense parameters from the STE-modified chain rules. As Eq. 3 shows, $g(\widetilde{\mathcal{W}}_t)$ is a rough estimate of the gradients for $\mathcal{W}_t$ due to the mismatch between the forward and backward passes. When conducting gradient descent to $\mathcal{W}$ with the rough gradients estimated by STE, discrepancies between the accurate gradients and approximated ones may lead to erroneous parameter updates. These imprecise value updates on the pruned parameters may further produce unstable alternations of the architecture of the on-the-fly pruned sparse neural networks in the forward pass, which causes the notable performance drop. To demonstrate the possible relationship between sparse network architecture updates and performance drops, we define SAD (Sparse Architecture Divergence) and measure the network architecture change with this metric.

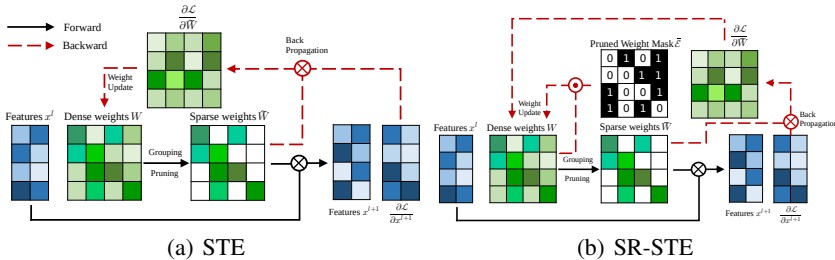

(a) STE
(b) SR-STE

Figure 2: In this figure, $\odot$ represents element-wise multiplication and $\otimes$ indicates matrix multiplication. (a) This figure shows the forward and backward pass during training an *N:M* sparse network. In the forward stage, $\widetilde{\mathcal{W}}$ is obtained by pruning $\mathcal{W}$. And in the backward stage, the gradient w.r.t. $\widetilde{\mathcal{W}}$ will be applied to $\mathcal{W}$ directly. (b) This figure illustrates the training process with SR-STE. The forward pass is the same as in (b). However, in the backward pass, the weights of $\mathcal{W}$ are updated by not only $\frac{\partial \mathcal{L}}{\partial \widetilde{\mathcal{W}}}$, but also $\bar{\mathcal{E}} \odot \mathcal{W}$, where $\bar{\mathcal{E}}$ is the mask matrix for the pruned weights in $\widetilde{\mathcal{W}}$.

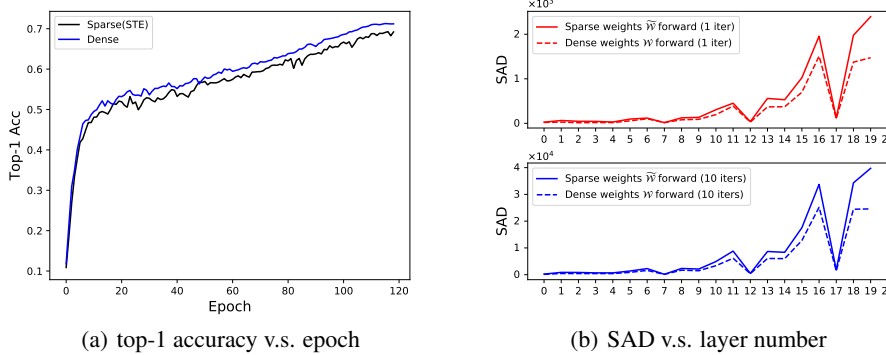

(a) top-1 accuracy v.s. epoch
(b) SAD v.s. layer number

Figure 3: We compare two networks respectively trained with regular SGD method and STE-modified gradient descent. (a) This figure shows sparse networks trained with STE has a significant performance drop in top-1 accuracy compared with dense networks. (b) This figure illustrates the layer-wise SAD between the weights after certain number of iterations and the initial weights, for two networks trained with STE (sparse forward) and regular SGD(dense forward). Compared with networks trained with sparse forward gradient, the one with dense forward gradient displays smaller SAD, indicating fewer updates in its sparse network architectures.

Before formally defining SAD, we first define the binary parameter *mask* produced in the magnitude-based pruning process as $\mathcal{E} = \{\mathcal{E}^l \in \{0,1\}^{N_l} : 0 < l \leq L\}$ where $N_l$ represents the length of $\mathcal{W}^l$. Specifically, if the $i$-th parameter of $\mathcal{W}^l$ survived (not pruned) in the pruned sub-network $\widetilde{\mathcal{W}}$, we set $\mathcal{E}_i^l = 1$, and $\mathcal{E}_i^l = 0$ otherwise. Thus, the sparse sub-network can be represented as $\widetilde{\mathcal{W}} = \{\mathcal{W}^l \odot \mathcal{E}^l : 0 < l \leq L\}$, where $\odot$ represents element-wise multiplication. For convenience, we define $\bar{\mathcal{E}} = \mathbf{1} - \mathcal{E}$.

For a single training run, we propose *Sparse Architecture Divergence* (SAD) to measure the change of the binary mask $\mathcal{E}$ from $W_i$ (the weights after the $i$-th iteration) to $W_j$ (the weights after the $j$-th iteration). We define $\text{SAD}_{i:j} = \|\mathcal{E}_j - \mathcal{E}_i\|_1$, where $\mathcal{E}_i$ and $\mathcal{E}_j$ are the binary masks for $W_i$ and $W_j$ respectively. This formula measures the number of connections(weights) that are pruned in $\widetilde{\mathcal{W}_i} = \mathcal{W}_i \odot \mathcal{E}_i$ and not pruned in $\widetilde{\mathcal{W}_j} = \mathcal{W}_j \odot \mathcal{E}_j$, or pruned in $\widetilde{\mathcal{W}_i}$ while not pruned in $\widetilde{\mathcal{W}_j}$. Smaller $\text{SAD}_{i:j}$ indicates less discrepancy between the network architectures of $\widetilde{\mathcal{W}_i}$ and $\widetilde{\mathcal{W}_j}$.

To reveal the impact of STE to the obtained sparse network architecture, we analyze SAD between the weights after different number of training iterations of two training schemes. Our quantitative results are shown in Fig. 3(b). The first scheme applies forward pass with dense weights and updates the net weights by $\mathcal{W}_{t+1} \leftarrow \mathcal{W}_t - \gamma_t g(\mathcal{W}_t)$. Then the corresponding sparse sub-network is obtained by pruning the trained dense network. The other updating scheme carries out forward pass with sparse layers and uses STE-modified chain rule, $\mathcal{W}_{t+1} \leftarrow \mathcal{W}_t - \gamma_t g(\widetilde{\mathcal{W}_t})$, in backward pass. We

define $^D\widetilde{\mathcal{W}}_i$ to be the sparse model pruned from the model trained for $i$ iterations of regular gradient descent, and $^S\widetilde{\mathcal{W}}_i$ to be the sparse model pruned from the network trained with $i$ iterations of STE-modified gradient descent. Let $^S\mathrm{SAD}_{0:t}^l$ denote SAD between the $l$-th layer of $^S\widetilde{\mathcal{W}}_0$ and $^S\widetilde{\mathcal{W}}_t$ trained with sparse forward. Similarly, $^D\mathrm{SAD}_{0:t}^l$ represents the SAD value between the $l$-th layer of $^D\widetilde{\mathcal{W}}_0$ and $^D\widetilde{\mathcal{W}}_t$ when trained with dense forward. As depicted in Fig. 3(b), for each layer number $l$, $^D\mathrm{SAD}_{0:t}^l < {}^S\mathrm{SAD}_{0:t}^l$ holds both when $t = 1$ and $t = 10$. Meanwhile, $|^S\mathrm{SAD}_{0:t}^l - {}^D\mathrm{SAD}_{0:t}^l|$ grows as $t$ increases from 1 to 10. This phenomenon indicates a positive correlation between the poor performance of a sparse neural network and high SAD.

Before our defined SAD, (Liu et al., 2020) proposes Neural Network Sparse Topology Distance (NNSTD) to measure topological distances between two sparse neural networks. It is worth noting that NNSTD reorders the neurons in each layer, based on their connections to the previous layer, to maximize the similarities between the two compared networks' topologies. Hence, NNSTD can only measure general topological differences between two networks but fails to reflect the transitions of individual connections' states (pruned or not pruned). However, when calculating SAD, the mask for each connection is directly computed without neurons being ordered, hence SAD could provide more precise estimation of actual state changes (from being pruned to not pruned, and vice versa) of network connections, which is what we most concern about in this paper. It is also worth noting that SAD has been implicitly adopted in existing published research papers. In RigL (Evci et al., 2019a), the authors consider the case of $\mathrm{SAD}_{t-1:t} = 2k$, where $k$ is dynamically calculated for each layer during training. RigL can achieve state-of-the-art results on training unsturctured sparse networks from scratch. Another example is that, when we prune the network at initialization and don't update the connections, according to the recent work (Frankle et al., 2020), the performance drops significantly. This can be regarded as setting $\mathrm{SAD}_{t-1:t} = 0$ during the whole training phase.

### 3.3 SPARSE-REFINED STRAIGHT-THROUGH ESTIMATOR (SR-STE)

Inspired by above observations made from SAD analysis, we aim to reduce SAD to improve the sparse network's performance. Since the magnitude of parameter $|w|$ are used as a metric to prune the network weights, we need to alternate the weight updating process in order to prevent high SAD. Two choices are left to us to achieve this goal: (1) restricting the values of weights pruned in $\widetilde{\mathcal{W}}_t$, (2) promoting the non-pruned weights in $\widetilde{\mathcal{W}}_t$. It is worth noting that although gradients of parameters calculated from STE are all approximated, the pruned parameters' gradients are more coarse than the non-pruned ones. Because for pruned parameters, the values to compute gradients in $\widetilde{\mathcal{W}}_t$ and the values to be updated in $\mathcal{W}_t$ are distinct, however for non-pruned parameters those two stages use the same values. These statements make penalizing weights pruned in $\widetilde{\mathcal{W}}_t$ a natural choice for restraining SAD.

Hence, we propose a sparse-refined term in the STE update formulation. We denote this new scheme as SR-STE. Compared to STE, when utilizing SR-STE in $N{:}M$ sparse model training, the backward pass is carried out with a refined gradients for pruned weights, as illustrated in Fig. 2(b). The purpose of the regularization term is to decrease the magnitude of the pruned weights, which are determined in the forward pass. Intuitively, we encourage the pruned weights at the current iteration to be pruned also in the following iterations so that the sparse architecture is stabilized for enhancing the training efficiency and effectiveness.

Formally, the network parameter update rule changes from Eq. 3 to the following formulation with a sparse-refined regularization term,

$$\mathcal{W}_{t+1} = \mathcal{W}_t - \gamma_t(g(\widetilde{\mathcal{W}}_t) + \lambda_W(\bar{\mathcal{E}}_t \odot \mathcal{W}_t)), \tag{4}$$

where $\bar{\mathcal{E}}_t$ denotes the mask for the pruned weights, $\odot$ denotes Hadamard product, $\lambda_W$ denotes the relative weight for the sparse-refined term, and $\gamma_t$ denotes the learning rate.

When $\lambda_W = 0$, Eq. 4 is equivalent to Eq. 3, which is the STE update rule. In general, we set $\lambda_W > 0$. SR-STE terms with a positive $\lambda_W$ set a constraint on and only on the pruned parameters, to prevent them from 1) being unpruned due to the different levels of mismatch between pruned parameter gradients and non-pruned parameter gradients, and 2) ineffectively alternating the pruned network architecture. When fewer sparse connections in the network are alternated, a more stable training process and a higher validation accuracy would be expected, which has been demonstrated in the analysis above and manifested in following experiments.

We perform extensive experiments with SR-STE, and these results can be found in Fig. 4. The experiments here are conducted with ResNet-18 (He et al., 2016) on ImageNet (Deng et al., 2009). In Fig. 4(a), 4 different settings of $\lambda_W$, namely $0, 0.0002, 0.00045, -0.00002$, are inspected. With $\lambda_W < 0$, the potential negative impact of pruned weights' coarse gradients are enlarged, which leads to the poorest top-1 accuracy (68.5%) and the most significant SAD. For smaller values of $\lambda_W$ corresponding to the standard version of SR-STE, SAD shrinks meanwhile the top-1 accuracy receives clear increase. Furthermore, we examine performances of three neural networks: 1) a dense neural network trained with regular SGD method; 2) an $N{:}M$ sparse network optimized with STE; 3) an $N{:}M$ sparse network trained with SR-STE. The curves of their top-1 accuracy for all training epochs are illustrated in Fig. 4(b). The accuracy curve of STE is consistently below the other two curves, and has more turbulence between different training epochs. Note that the SAD value is associated with the learning rate $\gamma$. For instance, SAD grows rapidly during the first 5 epochs in Fig. 4(a) since the learning rate increases from 0 to 0.1 in the so-called "warm-up" process. Besides, we also present other formations of sparse-refined term in Appendix A.5.

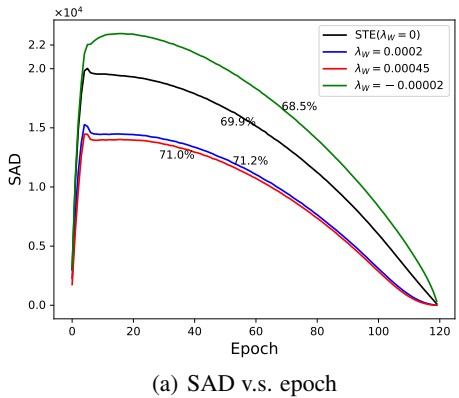
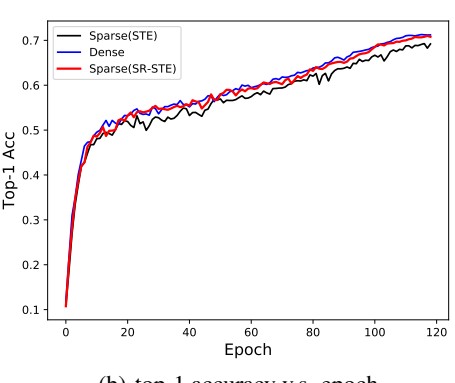

(a) SAD v.s. epoch  (b) top-1 accuracy v.s. epoch

Figure 4: (a) This figure illustrates SAD as a function of training epoch number with 4 different settings of $\lambda_W$ in the SR-STE term. When $\lambda_W < 0$, the perturbations brought by coarse gradients of sparse wights are widened, SAD gets higher and the top-1 accuracy becomes lower. When $\lambda_W$ is set to a reasonable positive value, sparse nets received high performance and low SAD. (b) This figure compares the top-1 accuracy curves of sparse net trained with STE, sparse net trained with SR-STE, and dense net. Sparse networks naively trained with STE have significant performance drop compared with dense ones. After introducing the SR-STE term into optimization process, the sparse network's performance jumps to a comparable level with dense networks.

## 4 EXPERIMENTS

In this section, we demonstrate the effectiveness of our proposed $N{:}M$ fine-grained structured sparsity neural network on computer vision tasks (*e.g.*, image classification, object detection, instance segmentation, and optical flow prediction) and machine translation tasks. For these experiments, the implementation details, including dataset settings, training schedules, and evaluation metrics, are listed in the Appendix A.4. Meanwhile, we set $\lambda_W$ as 0.0002 in all experiments because this value gives a good performance in our experiments.

### 4.1 IMAGE CLASSIFICATION

In this section, we first conduct several experiments to evaluate the effects of different $N{:}M$ sparse patterns and different training methods on the image classification benchmark ImageNet-1K (Deng et al., 2009) with different backbones. Then, we provide the performance of our proposed $N{:}M$ fine-grained structured sparsity network compared with the state-of-the-art sparsity methods.

**Different $N{:}M$ Sparse Patterns.** To investigate the performance of different $N{:}M$ sparse patterns, we exploit the popular ResNet50 (He et al., 2016) model with four different $N{:}M$ structural sparse patterns: 2:4, 4:8, 1:4, and 2:8. The baseline model is the traditional dense model. For the designed patterns, the 2:4 and 4:8 have the same sparsity 50%. The sparsity of 1:4 and 2:8 are both 75%. In Table 1, we observe that the 4:8 structural sparsity outperforms 2:4 with the same computational

Table 1: ImageNet validation accuracy on ResNet with different $N{:}M$ sparse patterns.

| Model | Method | Sparse Pattern | Top-1 Acc(%) | Params(M) | Flops(G) |
|---|---|---|---|---|---|
| ResNet50 | - | Dense | 77.3 | 25.6 | 4.09 |
| ResNet50 | SR-STE | 2:4 | 77.0 | 13.8 | 2.15 |
| ResNet50 | **SR-STE** | **4:8** | **77.4** | **13.8** | **2.15** |
| ResNet50 | SR-STE | 1:4 | 75.3 | 7.93 | 1.17 |
| ResNet50 | **SR-STE** | **2:8** | **76.2** | **7.93** | **1.17** |
| ResNet50 x1.25 | **SR-STE** | **2:8** | **77.5** | 11.8 | 1.79 |

Table 2: Experimental results of different training methods for training the $N{:}M$ sparse network.

| Model | Method | Sparse Pattern | Top-1 Acc | Epochs |
|---|---|---|---|---|
| ResNet18 | ASP (Nvidia, 2020) | 2:4 | 70.7 | 200 |
| ResNet18 | STE | 2:4 | 69.9 | 120 |
| ResNet18 | **SR-STE** | 2:4 | **71.2** | 120 |
| ResNet50 | ASP(Nvidia, 2020) | 2:4 | 76.8 | 200 |
| ResNet50 | STE | 2:4 | 76.4 | 120 |
| ResNet50 | **SR-STE** | **2:4** | **77.0** | **120** |

Table 3: Experimental results of the proposed $N{:}M$ sparse pattern with SR-STE and state-of-the-art sparsity methods. $^*$ imply that the first layer keep dense.

| Method | Top-1 Acc(%) | Sparsity(%) | Params(M) | Flops(G) | Structured | Uniform |
|---|---|---|---|---|---|---|
| ResNet50 | 77.3 | 0.0 | 25.6 | 4.09 | - | - |
| DSR$^*$ | 71.6 | 80 | 5.12 | 0.82 | ✗ | ✗ |
| RigL | 74.6 | 80 | 5.12 | 0.92 | ✗ | ✓ |
| GMP | 75.6 | 80 | 5.12 | 0.82 | ✗ | ✓ |
| STR | 76.1 | 81 | 5.22 | 0.71 | ✗ | ✗ |
| STE | 76.2 | 80 | 5.12 | 0.82 | ✗ | ✓ |
| **SR-STE** | **76.2** | **2:8** | **7.93** | **1.17** | ✓ | ✓ |
| RigL | 67.5 | 95 | 1.28 | 0.32 | ✗ | ✓ |
| GMP | 70.6 | 95 | 1.28 | 0.20 | ✗ | ✓ |
| STR | 70.2 | 95 | 1.24 | 0.16 | ✗ | ✗ |
| STE | 68.4 | 95 | 1.28 | 0.20 | ✗ | ✓ |
| **SR-STE** | **71.5** | **1:16** | **3.52** | **0.44** | ✓ | ✓ |

cost, and 2:8 also performs better than 1:4. the training curve in Fig. 6(a). It shows that with the same sparsity for $N{:}M$ structural sparse patterns, a larger $M$ will lead to better performance since it can provide more abundant convolution kernel shape (we visualize and analysis the convolution kernel shape in Appendix A.2). For the fixed $M$, we can adjust $N$ to obtain the different sparsity ranges. With the same $M$, it is reasonable that the performance of the larger $N$ is better due to more parameters and computational cost. Meanwhile, the performance of the ResNet50 with 1.25 width can achieve 77.5% top-1 accuracy about only 71% sparsity of original dense ResNet50. We also conduct several experiments with RegNetXs (Radosavovic et al., 2020) to evaluate the effectiveness of our proposed $N{:}M$ fine-grained structural sparse patterns in the compact models in Appendix A.3.

**Different Training Methods.** We also verify the effectiveness of the proposed SR-STE for training the $N{:}M$ sparse pattern neural network. In Table 2, we find that SR-STE outperforms the NVIDIA ASP method and STE with less training epochs and better accuracy.

**Comparison with State-of-the-arts.** Before the advent of $N{:}M$ fine-grained structured sparsity, there exist many state-of-the-art methods to generate sparsity models, including DSR (Mostafa & Wang, 2019), RigL (Evci et al., 2019a), GMP (Gale et al., 2019), and STR (Kusupati et al., 2020). SR-STE is compared to those methods on ResNet50 in mid-level sparsity( 80%) and ultra-level sparsity( 95%). Table 3 shows that SR-STE can outperform all state-of-the-art methods, even if other methods are unstructured sparsity. And STR (Kusupati et al., 2020) shows that training the model with non-uniform sparsity can improve the performance consistently, thus the SR-STE can extend the sparsity with non-uniform structural sparsity setting (*e.g.*, mixed $N{:}M$ fine-grained structural sparsity). We believe that the mixed $N{:}M$ sparsity could further improve the results and we leave this exploration for the future work.

## 4.2 OBJECT DETECTION AND INSTANCE SEGMENTATION

We further conduct experiments on the challenging COCO dataset (Lin et al., 2014) to evaluate the efficiency of the proposed approach for two important computer vision tasks, *i.e.*, object detection and instance segmentation. We exploit the classical model Faster RCNN (Ren et al., 2015) for object detection and Mask RCNN (He et al., 2017) for instance segmentation. All the experiments are conducted based on MMDetection (Chen et al., 2019). Table 4 and Table 5 show that 2:8 (25%) structured sparsity can achieve comparable result with dense baseline models. Furthermore, 4:8 (50%) sparsity can provide even better result than dense models. These results also illustrate that the $N$:$M$ sparsity pre-trained model gives a similar or better feature transfer ability.

Table 4: Object detection results on COCO.

| Model | Method | Sparse Pattern | LR Schd | mAP |
|---|---|---|---|---|
| F-RCNN-R50 | – | Dense | 1× | 37.4 |
| F-RCNN-R50 | **SR-STE** | 2:4 | 1× | **38.2** |
| F-RCNN-R50 | **SR-STE** | 2:8 | 1× | 37.2 |
| F-RCNN-R50 | – | Dense | 2× | 38.4 |
| F-RCNN-R50 | **SR-STE** | 2:4 | 2× | **39.2** |
| F-RCNN-R50 | **SR-STE** | 2:8 | 2× | 38.9 |

Table 5: Instance segmentation results on COCO.

| Model | Method | Sparse Pattern | LR Schd | Box mAP | Mask mAP |
|---|---|---|---|---|---|
| M-RCNN-R50 | – | Dense | 1× | 38.2 | 34.7 |
| M-RCNN-R50 | **SR-STE** | 2:4 | 1× | **39.0** | **35.3** |
| M-RCNN-R50 | **SR-STE** | 2:8 | 1× | 37.6 | 33.9 |
| M-RCNN-R50 | – | Dense | 2× | 39.4 | 35.4 |
| M-RCNN-R50 | **SR-STE** | 2:4 | 2× | **39.8** | **35.9** |
| M-RCNN-R50 | **SR-STE** | 2:8 | 2× | 39.4 | 35.4 |

## 4.3 OPTICAL FLOW AND MACHINE TRANSLATION

Optical flow prediction is one representative dense pixel-level prediction task in computer vision. We verify our proposed method on a recent state-of-the-art model RAFT (Teed & Deng, 2020) model on FlyingChairs (Dosovitskiy et al., 2015). The smaller value of the metric end-point-error (EPE) represents better performance. Compared with the dense model for optical flow prediction, Table 6 shows that our proposed method can achieve comparable accuracy with half of the parameters.

Besides the computer vision tasks, we investigate the effectiveness of our method on one of the most common tasks in natural language processing, *i.e.*, machine translation. We conduct our experiments based on Transformer, which employs a number of linear layers. We train our models with *transformer_base* adopted by (Vaswani et al., 2017), which contains a 6-layer encoder and a 6-layer decoder with 512-dimensional hidden representations. The larger value of the metric BLUE represents better performance. Compared with the dense model, Table 7 shows that our proposed method can achieve the negligible accuracy loss.

Table 6: RAFT results on FlyingChairs.

| Model | Method | Sparse Pattern | EPE | Params(M) | Flops(G) |
|---|---|---|---|---|---|
| RAFT | - | Dense | 0.86 | 5.3 | 134 |
| RAFT | **SR-STE** | **2:4** | **0.88** | **2.65** | **67** |

Table 7: MT Results on the EN-DE WMT'14.

| Model | Method | Sparse pattern | BLUE | Params(M) | Flops(G) |
|---|---|---|---|---|---|
| Transformer | - | Dense | 27.31 | 63 | 10.2 |
| Transformer | **SR-STE** | **2:4** | **27.23** | **31.5** | **5.1** |

## 5 DISCUSSION AND CONCLUSION

In this work, we present SR-STE for the first time to train $N$:$M$ fine-grained structural sparse networks from scratch. SR-STE extends Straight-Through Estimator with a regularization term to alleviate ineffective sparse architecture updates brought by coarse gradients computed by STE-modified chain rules. We define a metric, *Sparse Architecture Difference* (SAD), to analyze these architecture updates, and experimental results show SAD correlates strongly with pruned network's performance. We hope this work could shed light on machine learning acceleration and SAD could inspire more theoretical and empirical studies in sparse network training.

**Acknowledgements** We thank all the anonymous reviewers, Qingpeng Zhu and Tim Tsz-Kit LAU for their feedback on this work. This work is supported in part by SenseTime Group Limited, in part by Centre for Perceptual and Interactive Intellgience (CPII) Limited, in part by the General Research Fund through the Research Grants Council of Hong Kong under Grants CUHK 14207319/14208417.

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

# A APPENDIX

## A.1 ALGORITHM

---

**Algorithm 1** Training N:M sparse Neural Networks from Scratch of SR-STE

---

**Require:** $N$, $M$, $\lambda_W$, dataset $\mathcal{D}$,
 1: randomly initilized model $\mathcal{W}$,
 2: learning rate $\gamma_t$
 3: **for** each training iteration $t$ **do**
 4:     Sample mini batch of data$(\mathcal{X}, \mathcal{Y}) \sim \mathcal{D}$
 5:     $\widetilde{\mathcal{W}} \leftarrow S(\mathcal{W}, N, M)$                                                              ▷ Eq. 2
 6:     Obtain corresponding mask $\mathcal{E}$
 7:     $\ell \leftarrow \mathcal{L}(\widetilde{\mathcal{W}}, (\widetilde{\mathcal{X}}, \mathcal{Y}))$                          ▷ Forward pass
 8:     $g(\widetilde{\mathcal{W}}) = \frac{\partial \ell}{\partial \widetilde{\mathcal{w}}}$                                    ▷ Backward pass
 9:     $\mathcal{W} \leftarrow \mathcal{W} - \gamma_t g((\widetilde{\mathcal{W}}) + \lambda_W(\bar{\mathcal{E}} \odot \mathcal{W}))$,                      ▷ Eq. 4

---

## A.2 KERNEL SHAPE

Fig. 5 illustrates six learnt convolution kernels which are picked up from the trained ResNet50 2:8 sparse model. Note that, for these six convolution kernels, the shape of non-zero elements under the 2:8 sparsity constraints cannot be acquired or learnt in the case of 1:4 sparsity.

## A.3 REGNETXS ON IMAGENET-1K

We further verify whether *N:M* sparsity can boost the compact models. Recent RegNetXs (Radosavovic et al., 2020) are state-of-the-art and hardware-friendly models, which are the best models out of a search space with $10^{18}$ candidate models. The Table 8 shows that SR-STE can improve RegNetX002 performance than STE significantly, and the 2:4 structured sparsity can outperform

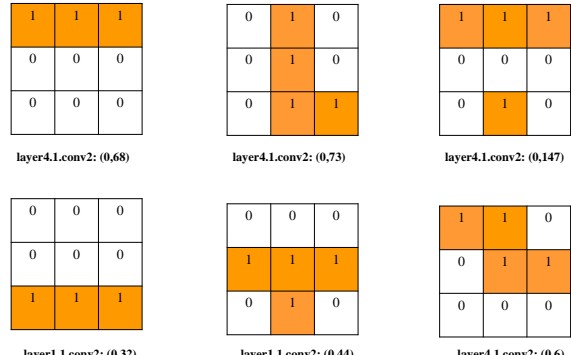

Figure 5: Illustration of kernel shape in ResNet50 with 2:8 structured sparsity trained model, layer1.1.conv2: (0,32) denotes layer name: (index of input channel, index of output channel).

the dense RegNetXs model with the same Flops. Therefore, we can obtain *N:M* fine-grained structured sparse model with SR-STE easily and we believe that the proposed *N:M* fine-grained structured sparsity method could become a standard technique on model deployment.

Table 8: ImageNet validation accuracy on RegNet with different $N$:$M$ sparse patterns.

| Model | Method | Sparse Pattern | Top-1 acc(%) | Flops(G) |
|---|---|---|---|---|
| RegNetX002 | **SR-STE** | 2:4 | **66.7** | 0.1 |
| RegNetX004 | **SR-STE** | 2:4 | **71.4** | 0.2 |
| RegNetX006 | **SR-STE** | 2:4 | **72.8** | 0.3 |
| RegNetX008 | **SR-STE** | 2:4 | **74.1** | 0.4 |
| RegNetX016 | **SR-STE** | 2:4 | **76.7** | 0.8 |
| RegNetX032 | **SR-STE** | 2:4 | **77.8** | 1.6 |
| RegNetX002 | – | Dense | 68.3 | 0.2 |
| RegNetX004 | – | Dense | 72.3 | 0.4 |
| RegNetX006 | – | Dense | 73.8 | 0.6 |
| RegNetX008 | – | Dense | 74.9 | 0.8 |
| RegNetX016 | – | Dense | 76.8 | 1.6 |

## A.4 IMPLEMENTATION DETAILS

### A.4.1 CLASSIFICATION

**Dataset.** ImageNet-1K (Deng et al., 2009) is a large-scale classification task, which is known as the most challenging image classification benchmark. ImageNet-1K dataset has about 1.2 million training images and 50 thousand validation images. Each image is annotated as one of 1000 object classes.

**Training scheduler.** All ImageNet-1K experiments trained on images of $224 \times 224$, the dense models baselines following the hyperparameter setting in (He et al., 2019). Specifically, all models are trained with batch size of 256 over 120 epochs and learning rates are annealed from 0.1 to 0 with a cosine scheduler and first 5 epochs the learning rate linearly increases from 0 to 0.1.

**Evaluation Metric.** All reported results follow standard Top-1 accuracy.

### A.4.2 OBJECT DETECTION AND INSTANCE SEGMENTATION

**Dataset.** All experiments are performed on the challenging MS COCO 2017 dataset (Lin et al., 2014) of 80 categories. It consists of $115K$ images for training (*train-2017*) and $5K$ images for validation (*val-2017*). We train models on the training dataset *train-2017*, and evaluate models on the validation dataset *val-2017*.

**Training scheduler.** For object detection and instance segmentation, we conduct these experiments on MMDetection (Chen et al., 2019). The 1x and 2x training schedule settings follow the settings in MMDetection (Chen et al., 2019).

**Evaluation Metric.** All reported results follow standard COCO-style Average Precision (AP) metric, *i.e.*, mAP of IoUs from 0.5 to 0.95.

### A.4.3 OPTICAL FLOW

**Dataset.** The optical flow prediction is conducted on the FlyingChairs (Dosovitskiy et al., 2015) dataset, which is a synthetic dataset with optical flow ground truth. This dataset consists of 22,872 image pairs and corresponding flow fields. The training dataset contains 22,232 samples and the validation dataset contains 640 test samples. We train the RAFT (Teed & Deng, 2020) model on the training dataset and report the final results on this validation dataset.

**Training scheduler.** We employ the original standard open framework[2] to run the RAFT (Teed & Deng, 2020) model. And the training settings for the FlyingChairs (Dosovitskiy et al., 2015) dataset have been listed in Teed & Deng (2020).

**Evaluation Metric.** We choose the endpoint error (EPE) to evaluate the predicted result. EPE is the Euclidean distance between the predicted flow vector and the ground truth, averaged over all pixels.

### A.4.4 MACHINE TRANSLATION

**Dataset.** For English-German translation, the training set consists of about 4.5 million bilingual sentence pairs from WMT 2014. We use newstest2013 as the validation set and newstest2014 as the test set. Sentences are encoded using BPE, which has a shared vocabulary of about 37,000 tokens.

**Training scheduler.** We use subword method (Sennrich et al., 2016) to encode source side sentences and the combination of target side sentences. The vocabulary size is 37,000 for both sides. Each mini-batch on one GPU contains a set of sentence pairs with roughly 4,096 source and 4,096 target tokens. We use Adam optimizer (Kingma & Ba, 2015) with $\beta_1 = 0.9$ and $\beta_2 = 0.98$. For our model, we train for 300,000 steps. We employ four Titan XP GPUs to train both the baseline and our model.

**Evaluation Metric.** All reported results follow standard BLUE (Papineni et al., 2002) on tokenized, truecase output.

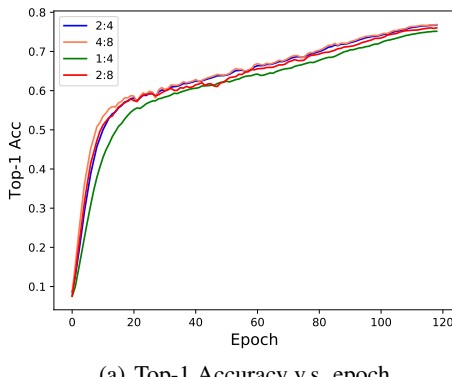

(a) Top-1 Accuracy v.s. epoch

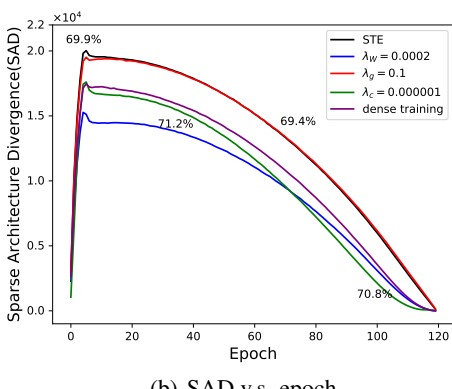

(b) SAD v.s. epoch

Figure 6: (a) This figure compares the top-1 accuracy curves of different *N:M* patterns with SR-STE. (b) This figure illustrates the SAD curves as each training epoch number with 3 different sparse-refined formulation

### A.5 OTHER REFINED FORMULATION

we present another sparse-refined regularization term, *i.e.*, sign constant, as follow:

$$\mathcal{W}_{t+1} = \mathcal{W}_t - \gamma_t(g(\widetilde{\mathcal{W}}_t) + \lambda_c(\bar{\mathcal{E}}_t \odot \text{sign}(\mathcal{W}_t))), \tag{5}$$

---

[2]https://github.com/princeton-vl/RAFT/

Apart from the parameter of model, we modified the Eq. 4 to apply on approximated gradient directly, as follow:

$$\mathcal{W}_{t+1} = \mathcal{W}_t - \gamma_t(g(\widetilde{\mathcal{W}_t}) + \lambda_g(\bar{\mathcal{E}}_t \odot (\gamma_t g(\mathcal{W}_t)))), \tag{6}$$

Fig. 6(b) depicts the curve of SAD of Eq. 4, Eq. 5, and Eq. 6. The refined term on gradients' SAD value is smaller than the STE in early stage of training while larger in the late training period, leading to worse performance (69.4%) than the performance of STE (69.9%). The sign constant SAD value converges similar to the Eq. 4, which improves the performance about 0.9%.

