# OpenReview forum: "Learning N:M  Fine-grained Structured Sparse Neural Networks From Scratch"
_ICLR.cc/2021/Conference — ICLR 2021 Poster_

### Official Review · AnonReviewer4 · 2020-10-27
**Clear motivation and detailed explanations on SAD, but I have concerns on structured sparsity and performance improvement**

**Rating:** 6
**Confidence:** 3

**Review:**

To maintain advantages of both unstructured and structured sparsity, the paper improves previous STE method based on the proposed SAD. The experiments on five types of applications show its outstanding performance.

pros.
1. The motivation of this paper, maintaining advantages of both unstructured fine-grained sparsity and structured coarse-grained sparsity, is clear and persuasive.
2. To validate outstanding performance of the proposed method, the authors did experiments on five different types of applications: image classification, object detection, instance segmentation, optical flow, and machine translation.
3. The authors provide detailed explanations on physical meaning of their proposed SAD.

cons.
1. In Section Introduction, I think the authors should formally explain meaning of "N:M" when "N:M" is mentioned for the first time.
2. In your github code train_imagenet.py, implementation of "devkit" package on 19th/20th/22nd line can not be found. I think your proposed method is mainly implemented in devkit.sparse_optimizer, while you do not include the most important code.
3. In Figure 1, physical meaning of variables R and C is not introduced.
4. The original STE is used for unstructured sparsity, while you improved STE for N:M structured sparsity. Why does your improvement encourage structured sparsity?
5. According to Table 2, the classification performance gap between STE and SR-STE is minor. So I think SR-STE is not a significant improvement compared to the original STE.  I am not 100% sure about my opinion on performance, since comparison between STE and SR-STE is not provided for the experiments on other applications (detection, segmentation, optical flow, and machine translation).

---

> ### Author Response · Authors · 2020-11-22
> **Responses to AnonReviewer4 (Part 2/2)**
>
> **Q5**: According to Table 2, the classification performance gap between STE and SR-STE is minor. So I think SR-STE is not a significant improvement compared to the original STE. I am not 100% sure about my opinion on performance, since a comparison between STE and SR-STE is not provided for the experiments on other applications (detection, segmentation, optical flow, and machine translation.
>
>
> **A5**: Regarding your concern, we explain the proposed SR-STE's performance in three parts:
>
> (i) For heavy models such as ResNet-50, the absolute performance gain is indeed "minor" but it is also significant if we consider it relatively. Specifically, if we compare the performance gain ($ACC_{SR-STE}-ACC_{STE}$) of SR-STE ($ACC_{dense}-ACC_{STE}$), it can be noticed that SR-STE significantly mitigates the performance drop of DNNs trained with STE from -0.9% to -0.3%. SR-STE can significantly mitigate 66.7% of the performance drop brought by STE! We believe the redundancy of ResNet-50 causes the *minor* performance drop of ResNet-50-STE and leaves SR-STE little space to improve over it.
>
>
> (ii) As for smaller-scaled models such as ResNet-18 and RegNet, our proposed SR-STE consistently improves the performance both relatively and absolutely, compared with STE. ResNet-18-SR-STE outperforms ResNet-18-STE by **1.3%** and at the same time mitigates the ResNet-18-STE's performance drop by 92.9% (from -1.4% to -0.1%). On the other hand, RegNetX002-SR-STE outperforms RegNetX002-STE by **3.7%**(see Table 8 in Appendix A.4).
>
> All the results in i) and ii) are demonstrated in the following table. Here, absolute improvement indicates $ACC_{SR-STE}-ACC_{STE}$ and relative improvement means $\frac{ACC_{SR-STE}-ACC_{STE}}{ACC_{dense}-ACC_{STE}}$.
>
> |  | Dense Top-1 Acc (%) | STE Top-1 Acc (%) | SR-STE Top-1 Acc (%) | Absolute Improvement (%) | Relative Improvement |
> |-|-|-|-|-|-|
> | RegNetX-002 | 68.3 | 63.0 | 66.7 | **3.7** | 69.8% |
> | ResNet-18 | 71.3 | 69.9 | 71.2 | 1.3 | 92.9% |
> | ResNet-50 | 77.3 | 76.4 | 77.0 | 0.6 | 66.7% |
>
>
>
> (iii) Following your suggestions, we have added **detection, segmentation, transformer, optical flow** performance comparisons in Tables 4, 5, 6, 7. From these results, **SR-STE** can outperform **STE** in all tasks.
>
>
> *Hope our above responses are helpful to address your concerns. If you have further questions, please let us know. Thanks!*

---

> ### Author Response · Authors · 2020-11-22
> **Responses to AnonReviewer4 (Part 1/2)**
>
> We highly appreciate your elaborate comments. We hope our answers below can address your concerns.
>
> **Q1**: Explanation of “N:M” in the introduction part.
>
> **A1**: We have given the explanation of "N:M" in the introduction (last line on page1)
>
> **Q2**: Codes cannot run.
>
> **A2**: We did include our implementations of SR-STE in the code. However, we didn't update train_imagenet.py after reconstructing our source codes, You can refer to our implementations of SR-STE from the 19th/20th/22nd line of https://github.com/anonymous-NM-sparsity/NM-sparsity/tree/main/devkit/sparse_ops. We also have updated train_imagenet.py correspondingly.
>
> **Q3**: Explanations of R and C under Fig. 1.
>
> **A3**: Thank you for your careful review. We have added explanations of these two notations in the figure description.
>
> **Q4**: The original STE is used for unstructured sparsity, while you improved STE for N:M structured sparsity. Why does your improvement encourage structured sparsity?
>
>
> **A4**: Before our proposal of N:M fine-grained structured sparsity, there are mainly two kinds of sparsity studied: fine-grained unstructured sparsity and coarse-grained structured sparsity sparsity. We agree with you that STE is mainly used for unstructured sparsity before our work but we think it is not because unstrucutred sparsity is "unstructured", instead, it is because the pruning metric in unstructured sparsity is **fine-grained**, so that STE can be applied to. And, in our work, although N:M sparsity is of the type of structured sparsity, it is also fine-grained, which means STE is a natural and suitable option in the process of training an N:M sparse DNN.
>
> On the other hand, although we mainly focus on N:M fine-grained structured sparsity here, we also perform some experiments of SR-STE on unstructured sparse DNNs for ablation study. In our results, SR-STE consistently outperforms STE in training both structured and **unstructured sparse** DNNs, which was not included in our first ICLR version since we think it deviates from our targeted problem. However, the question raised by you made us aware of the importance of this ablation study, so we re-added the results to our updated draft (In Table 3), detail as follows:
>
> |unstrucutred sparse DNNs  |        | Top-1 Acc (%) | Sparsity (%) | Improvement (%) |
> |-|:-|:-:|:-:|:-:|
> | ResNet18 | STE    |     66.3     |      90     |                |
> | ResNet18 | SR-STE |     69.1     |      90     |      **+2.8**      |
> | ResNet18 | STE    |     63.1     |      95     |                |
> | ResNet18 | SR-STE |     66.3     |      95     |      **+3.2**      |
>
> | unstrucutred sparse DNNs |        | Top-1 Acc (%) | Sparsity (%) | Improvement (%) |
> |-|:-|:-:|:-:|:-:|
> | ResNet50 | STE    |     73.8     |      90     |                |
> | ResNet50 | SR-STE |     74.9     |      90     |      **+1.1**      |
> | ResNet50 | STE    |     68.8     |      95     |                |
> | ResNet50 | SR-STE |     72.4     |      95     |      **+3.6**      |
>
>
> Specifically, in the results, when training a 90% sparse (90% of all weights are zeros) **unstructured sparse** ResNet18 and ResNet50 on ImageNet, the network trained with STE has a top-1 accuracy of 66.3%, which is far lower than that of SR-STE (69.1%). Here are the experimental results when STE and SR-STE are utilized in unstructured sparse DNNs.
>
> Table 3 shows that SR-STE outperforms **SOTA unstructured sparsity** method STE by about 2.2% with 95% sparsity, which verifies that our method can improve the unstructured sparsity rather than only for N:M structured.

---

### Official Review · AnonReviewer2 · 2020-10-28
**In this paper, the authors propose to train N:M structured sparse neural networks from scratch.  To improve the effectiveness of the training process, a sparse-refined straight through estimator (SR-STE) is proposed based on vanilla STE. Finally, the authors define Sparse Architecture Divergence (SAD) to indicate the topology change of the sparse network. The novelty of the proposed SR-STE is not enough, and the defined SAD lacks relevant theoretical support.**

**Rating:** 5
**Confidence:** 5

**Review:**

In this paper, the authors propose to train N:M structured sparse neural networks from scratch.  To improve the effectiveness of the training process, a sparse-refined straight through estimator (SR-STE) is proposed based on vanilla STE. Finally, the authors define Sparse Architecture Divergence (SAD) to indicate the topology change of the sparse network. The novelty of the proposed SR-STE is not enough, and the defined SAD lacks relevant theoretical support. Despite the good performance, I am concerned about the technical novelty of this paper.
(1) The main contribution of the proposed SR-STE lies in a sparse-refined regularization term. It seems like a mitigation measure, which cannot completely eliminate the negative impact caused by the approximate measures taken in the STE.
(2) And in my opinion, the analysis of SAD lacks relevant theoretical support and is not very convincing. I expect more explanations for SAD.

There are still several issues that need to be addressed. Firstly, the layout of this paper is not very suitable, and it is not very convenient in the process of finding the corresponding figures. Secondly, there are several narrative errors in this papere, which need to be checked carefully, e.g., in the third-to-last line of Section 1, I think it should be 'we propose a sparse refined term to enhance the effectiveness on training the sparse neural networks from scratch'; in the second-to-last line of Section 3.2, it should be '\left | ^{S}\textrm{SAD}_{0:t}^{l}- ^{D}\textrm{SAD}_{0:t}^{l}\right |'.

---

> ### Author Response · Authors · 2020-11-22
> **Responses to AnonReviewer2 (Part 2/2)**
>
> **Q3**: And in my opinion, the analysis of SAD lacks relevant theoretical support and is not very convincing. I expect more explanations for SAD.
>
> **A3**: Yes, SAD is an empirical measurement. Although it does not have theoretical support yet, we have empricially shown its relevance to the sparse network's performance in our experiments in Fig. 3(a) and Fig. 6(b). Furthermore, we justify the proposed SAD in two folds:
>
> (i) SAD has been implicitly adopted in existing published research papers. Two examples are demonstrated here:
>
> 1. In RigL [4], the authors consider the case of  **$SAD_t = 2 f_{decay}(t;\alpha,T_{end})(1-s^{l})N^l$**, RigL can achieve SOTA results on training unsturctured sparse networks from scratch.
> 2. If we prune the network at initialization and don’t update the connections, then according to the recent work [2], the performance drops significantly [2]. This can be regarded as setting **$SAD_t=0$** during the whole training phase.
>
>
> (ii) To the best of our knowledge, we are the first to explicitly define and investigate SAD on how it relates to the sparse model's performance during sparse training. Our study about the relevance between SAD and the sparse model performance is shown in Fig. 3(a) of our paper. In Fig. 3(a), the curve that corresponds to a higher accuracy tends to consistently have a larger SAD value during the training process.
>
>
> [1] Ma, Xiaolong, et al, "Non-Structured DNN Weight Pruning – Is It Beneficial in Any Platform?,"" IEEE Transactions on Neural Networks and Learning Systems (TNNLS), 2020.
>
> [2] Frankle, Jonathan, et al. "Pruning Neural Networks at Initialization: Why are We Missing the Mark?." arXiv preprint arXiv:2009.08576 (2020).
>
> [3] Renda, Alex, Jonathan Frankle, and Michael Carbin. “Comparing rewinding and fine-tuning in neural network pruning.” arXiv preprint arXiv:2003.02389 (2020).
>
> [4] Evci, Utku, et al. "Rigging the lottery: Making all tickets winners." arXiv preprint arXiv:1911.11134 (2019).
>
> [5] Dettmers, Tim, and Luke Zettlemoyer. "Sparse networks from scratch: Faster training without losing performance." arXiv preprint arXiv:1907.04840 (2019).
>
> [6] Nvidia. "Nvidia  a100  tensor  core  GPU  architecture." https://www.nvidia.com/content/dam/en-zz/Solutions/Data-Center/nvidia-ampere-architecture-whitepaper.pdf, 2020.
>
>
> **Q4**: There are still several issues that need to be addressed.
>
> **A4**: Following your suggestions, we tried our best on revising and proof-reading of our manuscript.
>
> *Hope our above responses are helpful to address your concerns. If you have further questions, please let us know. Thanks!*

---

> ### Author Response · Authors · 2020-11-22
> **Responses to AnonReviewer2 (Part 1/2)**
>
> We sincerely appreciate your valuable comments on our work. To address your concerns, our responses are demonstrated below:
>
> **Q1**: It seems like a mitigation measure, which cannot completely eliminate the negative impact caused by the approximate measures taken in the STE.
>
> **A1**: Yes, the proposed SR-STE is a mitigation that cannot completely eliminate the negative impacts. However, we justify SR-STE's contributions from two aspects.
>
> (i) Our proposed SR-STE can significantly close the performance gap between N:M sparse networks and dense ones, compared with other methods such as STE and two-stage methods proposed by Nvidia [6]. For example, we can mitigate the performance drop of 2:4 sparse ResNet-18 brought by training with STE by 92.9% (refer to Q4 in our response to AnonReviewer4). Although we do not fully eliminate the negative impact brought by STE, we do regard SR-STE as an effective and versatile strategy for N:M sparsity training. We sincerely hope our work can contribute to the hardware-aided deep model acceleration and shed light on further research and investigations in N:M sparsity training.
>
> (ii) On the other hand, in a larger scope, we are the first to attempt to analyze and mitigate the negative impact brought by STE when it is applied to N:M sparsity training. Our experimental results show that 2:4 structured sparse model trained with SR-STE can achieve **comparable or even better** results with **negligible extra training cost** and **only a single easy-to-tune hyperparameter $\lambda_w$** than original dense models. SR-STE consistently performs well on **different tasks** (classification, detection, machine translation), **different model** architectures (ResNet18/50, RegNetXs, Faster RCNN, Mask RCNN, RAFT, Transformer), and **different sparsity types** (unstructured and N:M structured sparsity).
>
>
> **Q2**: The novelty of the proposed SR-STE is not enough.
>
> **A2**: We propose the first method to tackle the problem of how to train an N:M structured sparsity network. This is a novel problem that has never been investigated before. We also show that simply adopting STE cannot effectively train an N:M structured sparsity network. The proposed SR-STE has the advantages of being **simple** and **easy** when being applied to **different large-scale tasks**(classification, detection, segmentation, machine translation, optical flow) with different types of sparsity (structured, unstructured).
>
>
>
> Comparisons of top-1 accuracy between **SOTA** coarse-grained structured and N:M structured ResNet50 networks on ImageNet are shown below:
>
> |                           |                               |                                    |
> |-|-|-|
> | Coarse structured | 76.1 (compression ratio=1) [3] | 74.3 (compression ratio = 1.26) [3] |
> | N:M structured            | **77.4** (compression ratio = 1)   | **76.4** (compression ratio = **4**)       |
>
> Furthermore, in the paper, we demonstrate the rationality of SR-STE with careful discussions and adequate experiments, which makes SR-STE a universal and reliable recipe in training N:M sparse DNNs. Other than SR-STE, we also have two other major contributions:
>
> (i) We propose a more general form of structured fine-grained sparse DNNs that can be accelerated on designated hardware. It has huge advantages over the other two forms of sparse DNNs that have been studied, namely, unstructured sparse DNNs and coarse-grained sparse DNNs. The efficacy of unstructured sparsity is highly limited and should not be further studied according to [1]. As for the coarse-grained sparsity (channel/filter pruning), it has a much lower performance compared with sparse neural networks (76.1 (coarse-grained) vs 77.4 (N:M structured) with 50% sparsity, in the above Table) according to [3], and even more performance drop with increasing compression ratios. N:M sparsity's outstanding performance compared to all other existing sparsity methods proves its huge potential in deep model acceleration.
>
> (ii) We propose SAD, an important metric that is closely related to model performance during training sparse neural networks according to our investigations, to foster future research on measuring topological changes of sparse DNNs in the training stage.

---

### Official Review · AnonReviewer3 · 2020-10-28
**Simple technique, but nice results**

**Rating:** 6
**Confidence:** 3

**Review:**

The authors proposed a new method for training N:M fine-grained structured sparse networks from scratch. The authors found that the SAD metric, which measures the number of weights whose pruning state is changed, became higher if the existing STE is used to train sparse networks and this metric had the positive relationship with accuracy drop. To reduce the SAD, SR-STE which gives higher weight decay coefficient to the pruned weights is applied to train sparse networks, improving the accuracy of sparse networks trained from scratch.

SR-STE which is penalizing the pruned weights is somewhat simple, but the results in the paper are strong. The proposed method supports to increase the accuracy of N:M fine-grained structured sparse networks, and it can be applied in many diverse tasks regardless of pruning method like general unstructured pruning. I think it is a good technique to easily apply when pruning states of each weight are continuously changed.

However, weight-sorting process which occurs in every training step can decrease the training speed compared to other threshold-based pruning method. Could you compare the actual consuming time to train sparse network for each training method?

Comment: Below are some typos I think.

1. 3.2. Analysis: I think |^{S}{SAD}^{l}_{0:t}-^{D}{SAD}^{l}_{0:t}| instead of |^{D}{SAD}^{l}_{0:t}-^{D}{SAD}^{l}_{0:t}| is likely to be the right expression in context.
2. 3.2. Analysis: The sentence "This formula measures the number of neurons that are pruned~" is unfamiliar to me, because "neuron" is usually used when depicting activation, not weight of artificial neural network. I think "weight" is more matched expression.
3. The stated GFLOPs of ResNet-50 (2:8 sparse pattern, SR-STE method) are different in Table 1 (1.02) and Table 3 (0.1). I think the former is right.

---

> ### Author Response · Authors · 2020-11-22
> **Responses to AnonReviewer3**
>
> Thank you for your comments and appreciation of our work. To relieve some of your concerns, we have posted our responses below:
>
>
> **Q1**: However, weight-sorting process which occurs in every training step can decrease the training speed compared to other threshold-based pruning method. Could you compare the actual consuming time to train sparse network for each training method?
>
> **A1**: The weight-sorting process of N:M structured sparsity is very efficient, compared with other fine-grained pruning methods, such as the weight sorting method [1] and threshold-based pruning method [2]. Implementation details and efficiency benchmarks are shown as follows:
>
> |weight_sorting(N:M structured)(ours)   |    weight_sorting(unstructured)[1]    | threshold-based(unstructured)[2] |
> |:----------:|:-------:|:--------------:|
> | 0.00045s | 0.0021s    |     0.00041s     |
>
> (i) Our weight-sorting implementation of achieving N:M sparsity is shown below:
>
> ```python
> def forward(ctx, weight, N=4, M=2):
>     output = weight.clone()
>     length = weight.numel()
>     group = int(length/M)
>     weight_temp = weight.detach().abs().reshape(group, M)
>     index = torch.argsort(weight_temp, dim=1)[:, :int(M-N)]
>
>     # compute the mask ($epsilon_t$ in the paper)
>     mask = torch.ones(weight_temp.shape, device=weight_temp.device)
>     mask = mask.scatter_(dim=1, index=index, value=0).reshape(weight.shape)
>
>     return output*mask, mask
> ```
>
> The weight-sorting technique in our implementation is very computationally efficient during the training process. It only takes **0.00045s** (averaged over 1000 iterations with batch size 256), which takes only 0.37% of 0.122s (the average time cost of one iteration).
>
> (ii) Recent SOTA unstructured sparsity methods also have to sort the weight tensor, which takes approximately **0.0021s** in one iteration, over 3.6 times more than SR-STE. Compared to N:M sparsity, when computing argsort(), unstructured sparsity methods need to sort the weights in one layer as a whole, which is more time-consuming than N:M sparsity. Because in N:M sparsity, weights are sorted inside its N-sized groups, which can be paralleled in popular deep learning libraries. The forward implementation of unstrucutred sparsity with STE is shown as follows:
>
> ```python
> def forward(ctx, weight, pruning_ratio=0.5):
>
>     output = weight.clone()
>     length = weight.numel()
>     weight_temp = weight.detach().abs().reshape(length)
>     index = torch.argsort(weight_temp, dim=0)[:int(pruning_ratio*length)]
>     mask = torch.ones(weight_temp.shape, device=weight_temp.device)
>     mask = mask.scatter_(dim=0, index=index, value=0).reshape(weight.shape)
>
>     return output*mask, mask
> ```
>
> (iii) We also implement the threshold method [2], which takes **0.00041s** (comparable with SR-STE) in one iteration, the implementation details are shown as follows:
>
> ```python
> def sparse_thres(W, crate=1.0):
>     W_abs = W.detach().abs()
>     # compute the threshold according to [2]
>     threshold = 0.9*(torch.mean(W_abs)+crate*torch.std(W))
>     mask = (W_abs<threshold)
>
>     return W*mask
> ```
>
> Besides, we also provide complete training time. Compared to 20h 46m 28s, which is the time cost to train a dense ResNet-18 network from scratch, training a sparse one with SR-STE takes 21h 20m 15s, which only adds 2.7% extra cost.
>
> [1]. Renda, Alex, Jonathan Frankle, and Michael Carbin. "Comparing rewinding and fine-tuning in neural network pruning." arXiv preprint arXiv:2003.02389 (2020).
>
> [2]. Guo, Yiwen, Anbang Yao, and Yurong Chen. "Dynamic network surgery for efficient DNNs." Advances in neural information processing systems. 2016.
>
> **Q2**: Comment: Typos.
>
> **A2**: We have gone through the paper and fixed the typos in the paper.
>
>
> *Hope our above responses are helpful to address your concerns. If you have any questions, please let us know. Thanks!*

---

### Official Review · AnonReviewer1 · 2020-10-29
**SR-STE and happy SAD**

**Rating:** 6
**Confidence:** 4

**Review:**

Summary

The paper introduces a new sparse training algorithm (SR-STE) based on the straight through estimator which is specially designed for the hardware constraints of Nvidia A100 GPU. Auxiliary, in order to study better this algorithm, the paper introduces also a metric (SAD) to measure the changes in the sparse network topology during training. The contributions have real added value as they show that sparse neural networks can actually benefit of hardware designed to consider sparsity. The experiments on CNNs and Transformers support the claims.

Strong Points

•	SR-STE can take real advantage of A100 sparse capabilities.

•	The algorithm is designed for the general form of n:m sparsity and, likely, can be used also for the next generation of GPUs with sparse networks support.

•	It is the first study which, up to my knowledge, shows consistently that sparse networks can outperform dense networks also for convolutional layers.

•	The paper is written in a clear manner and well structured.

Weak points

•	The proposed SAD metric seems to work just on the same network during its own training process, as it does not consider the fact that at different training runs the hidden neurons may develop differently.

•	The related work is not complete

•	The paper has a number of sloppy written passages and inconsistencies in the mathematical formalism.

During the rebuttal, I would suggest to the authors to consider the following comments:

1) Improve the mathematical formalism and perform a careful proof-read to make sure that all notations are consistent. For instance page 5,  2nd paragraph, E_f is actually \epsilon_f?;  page 5, third paragraph, the last equation has two equal terms, and so on.

2) I believe that it would be informative to discuss the SAD metric limitations and advantages in comparison with other state-of-the-art metrics suitable to measure the distance between two sparse topologies, e.g. NNSTD https://arxiv.org/abs/2006.14085 .

3)  Related work. the 4th paragraph on page 2 about the one-stage scheme to obtain efficient sparse networks is missing the first two references from 2017 which introduced sparse training with dynamic topology: https://arxiv.org/abs/1707.04780, https://arxiv.org/abs/1711.05136 . Auxiliary, I believe that also this work has to be cited https://arxiv.org/abs/1907.04840 .

4) The last phrase of the above mentioned paragraph and the next one are, up to my knowledge, accurate just for CNNs. It has been shown already that for other types of neural networks (e.g. MLPs), sparse networks can easily and constantly outperform dense networks (e.g. https://arxiv.org/abs/1905.09397). Please make this discussion more accurate and informative.

5)  Please read the paper thoroughly to address all typos and small inconsistencies. E.g., In Tables 1 and 2 “ours” appear just in some cases of SR-STE; In Table 3, STR appears twice with exactly the same sparsity level, but different number of parameters (more clarifications would make easier the reader job).

---

> ### Author Response · Authors · 2020-11-22
> **Responses to AnonReviewer1**
>
> We sincerely thank you for your comments as well as the appreciation of our work. Our responses to your concerns are demonstrated below:
>
> **Q1**: The proposed SAD metric seems to work just on the same network during its own training process, as it does not consider the fact that at different training runs the hidden neurons may develop differently.
>
> **A1**: Yes, SAD only works on the same network, since we use SAD here only to analyze how the topology of the **same** neural network evolves in the training process with STE or SR-STE. The research of effective metrics to compare topology differences between individually trained networks will be left to future work.
>
>
>
> **Q2**: The related work is not complete
>
> **A2**: Due to the page limit of our initial ICLR submission, we did not add some related work in our first version. We totally agree with you and have added more related discussion in the revised version (2nd Paragraph Page 3).
>
> **Q3**: The paper has a number of sloppy written passages and inconsistencies in the mathematical formalism.
>
> **A3**: We have reviewed our paper regarding your proposed problems and have updated the draft.
>
> **Q4**: Comparisons with NNSTD (https://arxiv.org/pdf/2006.14085 )
>
> **A4**: Thanks for pointing out the interesting and related paper, we have added discussions about NNSTD at the end of 2nd Paragraph on Page 6.
>
> **Q5**: The last phrase of the above-mentioned paragraph (the 4th paragraph on page 2) and the next one are, up to my knowledge, accurate just for CNNs. It has been shown already that for other types of neural networks (e.g. MLPs), sparse networks can easily and constantly outperform dense networks (e.g. https://arxiv.org/abs/1905.09397). Please make this discussion more accurate and informative.
>
> **A5**: Thank you for your suggestions. We have added proper phrases to avoid readers leaving the discussion scope, which is about training sparse CNN only. We also added discussions on MLPs where sparse networks' performances may constantly surpass those of the dense ones.
>
> *Hope our above responses are helpful to address your concerns. If you have further questions, please let us know. Thanks!*

---

### Author Response · Authors · 2020-11-22
**Paper update**

Thanks to all the reviewers for their constructive suggestions and comments. We really appreciate all your inputs. In this updated version, our paper has been refined after we carefully considered all reviewers’ suggestions.

1. Related work is added in the 2nd paragraph on Page 3. (AnonReviewer1)
2. Discussions about MLPs are added in the 4th paragraph on Page 2. (AnonReviewer1)
3. The inconsistencies in Tables 1, 2, 3 are fixed. (AnonReviewer1)
4. Explanations of SAD are refined in 1st paragraph on Page 5. (AnonReviewer1)
5. Comparisons and discussions between SAD and NNSTD are added in the 2nd paragraph on Page 6. (AnonReviewer1)


6. The layout of our paper is refined, and the figures are rearranged so that readers can easily find corresponding information. (AnonReviewer2)
7. Typos in our paper are fixed in the last paragraph on Page 2 and 1st paragraph on Page 6. (AnonReviewer2)
8. Explanations of SAD and comparisons between SAD with other relevant methods are demonstrated in the 2nd paragraph on Page 6. (AnonReviewer2)



9. Typos in the demonstration of SAD are fixed in 1st paragraph on Page 6. (AnonReviewer3)
10. Typos in Tables 1 and 3 are fixed. (AnonReviewer3)

11. The explanation of N:M is added in the last line on Page 1. (AnonReviewer4)
12. Explanations of R and C are added in the caption of Fig. 1 (AnonReviewer4)
13. Experimental results on unstructured sparsity are added in Table 3. (AnonReviewer4)
14. Experimental results on more tasks are added in Tables 4, 5, 6, 7. (AnonReviewer4)

Apart from the issues addressed above, we also revised our paper with regard to consistency and readability.

---

### Decision · Program_Chairs · 2021-01-07
**Final Decision**

**Decision:**

Accept (Poster)

**Comment:**

Reviewers like the simplicity of the approach and the fact that it works well.